# Microbiological Survey of 47 Permanent Makeup Inks Available in the United States

**DOI:** 10.3390/microorganisms10040820

**Published:** 2022-04-15

**Authors:** Sunghyun Yoon, Sandeep Kondakala, Seong Won Nho, Mi Sun Moon, Mei Chiung J. Huang, Goran Periz, Ohgew Kweon, Seongjae Kim

**Affiliations:** 1Division of Microbiology, National Center for Toxicological Research, U.S. Food and Drug Administration, 3900 NCTR Road, Jefferson, AR 72079, USA; sunghyun.yoon@fda.hhs.gov (S.Y.); sandeep.kondakala@fda.hhs.gov (S.K.); seongwon.nho@fda.hhs.gov (S.W.N.); 2Office of Cosmetics and Colors, Center for Food Safety and Applied Nutrition, U.S. Food and Drug Administration, 5001 Campus Drive, College Park, MD 20740, USA; misun.moon@fda.hhs.gov (M.S.M.); jo.huang@fda.hhs.gov (M.C.J.H.); goran.periz@fda.hhs.gov (G.P.)

**Keywords:** permanent makeup (PMU) ink, microbial contamination, bacteria

## Abstract

In two previous surveys, the U.S. Food and Drug Administration (FDA) identified microbial contamination in 53 of 112 (47%) unopened tattoo inks and tattoo-ink-related products (e.g., diluents) from 15 manufacturers in the U.S. In this study, we primarily focused our microbiological survey on permanent makeup (PMU) inks. We conducted a survey of 47 unopened PMU inks from nine manufacturers and a comparative species-centric co-occurrence network (SCN) analysis using the survey results. Aerobic plate count and enrichment culture methods using the FDA’s *Bacteriological Analytical Manual* (BAM) Chapter 23 revealed that 9 (19%) inks out of 47, from five manufacturers, were contaminated with microorganisms. The level of microbial contamination was less than 250 CFU/g in eight inks and 980 CFU/g in one ink. We identified 26 bacteria that belong to nine genera and 21 species, including some clinically relevant species, such as *Alloiococcus otitis*, *Dermacoccus nishinomiyaensis*, *Kocuria rosea*, and *Pasteurella canis*. Among the identified microorganisms, the SCN analysis revealed dominance and a strong co-occurrence relation of spore-forming extreme environment survivors, *Bacillus* spp., with close phylogenetic/phenotypic relationships. These results provide practical insights into the possible microbial contamination factors and positive selection pressure of PMU inks.

## 1. Introduction

Permanent makeup (PMU) or micropigmentation is a type of tattoo [1]. Because PMU is often performed for medical or aesthetic purposes, most commonly in the facial area, it is also called a “cosmetic tattoo”, while traditional and decorative tattoos are often applied to other parts of the body [2,3]. As in tattooing, colored pigment is injected through a needle into the skin to produce designs that resemble makeup, such as eyeliner, lip liner, eyebrows, or other makeup [1,4]. Tattoo and PMU inks are regulated by the U.S. Food and Drug Administration (FDA) as cosmetics, and the pigments in the inks, which are considered color additives, are subject to premarket approval [5]

PMU, along with decorative tattoos, has become popular worldwide over the last several decades [2,6]. As PMU has become more prevalent, risks and complications associated with PMU have also increased [4,6,7]. Between 2003 and 2004, more than 150 cases of adverse reactions occurring in consumers associated with PMU were reported to the FDA [5]. While it is difficult to attribute any of the adverse events to a specific cause, it is likely that there are health risks from PMU inks contaminated with pathogenic microorganisms [8,9]. In recent years, a number of PMU ink recalls due to microbial contamination have occurred [5,6]. However, relatively little is known about the occurrence of microbial contamination in commercially available PMU inks, compared with tattoo inks, of which up to 86% have been shown to be contaminated with microorganisms, depending on the survey [10,11,12,13,14,15].

Our two prior surveys of tattoo inks and tattoo-ink-related products, including PMU inks and ink diluents, revealed that 53 out of 112 (47%) unopened and sealed tattoo and PMU inks from 15 manufacturers were contaminated with microorganisms [14,15]. Among those inks surveyed, 29 PMU inks (23 in the first and 6 in the second survey) from five manufacturers were analyzed, and 14 of them (48%) from four manufacturers were found to be microbially contaminated. While the rate of PMU ink contamination was consistent with that of tattoo ink contamination (51%, 39 out of 77 tattoo inks), the number of PMU inks and range of manufacturers tested are not enough to generalize the survey results and to understand factors that contribute to microbial contamination.

In this study, we surveyed only PMU inks, including microblading (Mb) inks, which are PMU inks used on eyebrows [7]. We conducted a microbiological survey of 47 PMU inks and combined the data with our two previous surveys in order to perform a network-based comparative analysis and to gain systematic insights into factors that may influence microbial contamination of PMU inks.

## 2. Materials and Methods

### 2.1. PMU Inks

PMU inks were purchased from 9 manufacturers in the U.S. during February and March of 2019. We purchased 2 to 6 bottles of each individual ink with the same lot number. Upon receipt of survey samples, we checked seal integrity and stored the inks at room temperature. We recorded ink product label information, including country of origin, manufacturer, distributor, lot number, ingredients, sterility claim, and expiration date.

### 2.2. Microbiological Analysis of PMU Inks

We analyzed PMU inks for bacterial and fungal contamination based on the methods described in the FDA’s *Bacteriological Analytical Manual* (BAM) Chapter 23 [16]. This chapter provides the agency’s preferred laboratory procedures for testing cosmetics for the presence and identity of microorganisms (https://www.fda.gov/food/laboratory-methods-food/bam-methods-cosmetics, accessed on 1 March 2022). Briefly, we serially diluted ink suspensions using modified letheen broth (MLB) and plated 1 mL of 10^−1^ dilution (500 µL × 2 plates) and 100 µL of each 10^−1^, 10^−2^, and 10^−3^ dilution on modified letheen agar (MLA) plates and Potato Dextrose agar (PDA) with chlorotetracycline (40 µg/mL) in duplicate, for detection of bacteria and fungi, respectively. Diluted samples were incubated for 7 days for enrichment and then streaked (~5 µL) on MLA plates to detect microbial contaminants. As positive and negative controls, plates and culture media, with and without spike of test microorganisms *Staphylococcus aureus* (ATCC 25923), *Candida albicans* (ATCC 10231), *Pseudomonas aeruginosa* (ATCC 27853), and *Klebsiella pneumoniae* (ATCC 13883), were tested. Contamination of nontuberculous mycobacteria (NTM) was also analyzed based on the method described by the Office of Regulatory Affairs/U.S. FDA. The NTM detection method was originally developed to isolate and identify NTM associated with tattoo-related outbreaks [17]. Microbial contaminants were identified using VITEK 2 Compact System (BioMérieux, Durharn, NC, USA.), with Gram-negative (GN), Gram-positive (GP), and *Bacillus* (BCL) colorimetric cards. Sequencing of 16S rRNA gene was also used to identify bacteria using primers 27f and 1492r [18]. PCR products were purified using ExoSAP-IT (USB Corporation, Cleveland, OH, USA), as recommended by the manufacturer. DNA sequencing was performed by a Core Facility at the University of Arkansas for Medical Sciences in Little Rock, Arkansas (http://mbim.uams.edu/research-cores/dna-sequencing-core-facility, accessed on 1 March 2022).

### 2.3. Species-Centric Co-Occurrence Network (SCN) Analysis

An SCN analysis was performed as previously described [14]. Briefly, using a custom python script, a species–sample matrix (i.e., a presence–absence matrix) was generated from the results of the previous and present survey, and then a co-occurrence matrix was generated from the presence–absence matrix. In a species-centric co-occurrence network, if two bacterial species exist in a sample, these two bacterial species are associated with each other and form a co-occurrence relation. In a species-centric co-occurrence network, the nodes represent bacterial species whose edges (i.e., connection degree) indicate a relationship between bacterial species. The node size and line width are weighted by bacterial isolate occurrence counts and frequency of co-occurrence, respectively. Gephi (https://gephi.org/, accessed on 1 March 2022) was used for network analysis and visualization.

## 3. Results and Discussion

### 3.1. Microbial Contamination of PMU Inks

We surveyed 47 unopened and sealed PMU inks from nine manufacturers (Table 1, Figure 1, and Appendix A). Aerobic plate count (APC) coupled with enrichment culture methods revealed that nine (19%) PMU inks from five manufacturers contained bacteria. Out of 47 surveyed inks, 18 (38%) from four manufactures were labeled as sterile. Out of the 18 inks which made claims of sterility, 3 inks (17%) from two manufacturers were found to contain microorganisms. The level of microbial contamination detected was <250 CFU/g in eight inks and 980 CFU/g in one ink.

In our previous two surveys, we found 14 out of 29 PMU inks (48%) contained microorganisms [14,15]. If we add up all the PMU samples from this survey and the previous ones, a total of 23 (9 + 14) out of 76 (47 + 29) PMU inks (30%) contained microorganisms. The rate of microbial contamination observed in the PMU inks was 30%, which is lower than that of tattoo inks, where 39 out of 77 tattoo inks (50.6%) had been found to be contaminated [14,15]. In addition, the concentration of microorganisms found in PMU was much lower than that found in tattoo inks. While the contamination level in 51% of the tattoo inks (20 inks out of 39 contaminated tattoo inks) was found to be higher than 10^3^ CFU/g, with the highest being over 10^8^ CFU/g, microbial concentrations in contaminated PMU inks were mostly <250 CFU/g, with only one being 980 CFU/g. Our results show that PMU inks are less likely to be contaminated with high levels of microorganisms than tattoo inks.

### 3.2. Identification of Microorganisms Isolated from PMU Inks

We identified a wide variety of bacteria using VITEK and 16S rRNA sequence analysis. As shown in Table 1, 26 bacterial isolates, belonging to nine genera and 21 species, were identified. Seven out of nine contaminated PMU inks contained multiple strains of bacteria. For example, sample #37 produced a growth of 12 different species of bacteria. Isolates of genus Bacillus were dominant with 15 unique species (58%). Identification included possible pathogenic bacteria, such as Alloiococcus otitis, Dermacoccus nishinomiyaensis, Kocuria rosea, and Pasteurella canis. Although the bacteria are involved in human infections [19,20,21,22], they have never been previously reported as tattoo ink contaminants.

### 3.3. Species-Centric Co-Occurrence Network (SCN) of the Bacterial Contaminants from PMU Inks

#### 3.3.1. Newly Identified 14 Species Belonging to 6 Genera as a Bacterial Contaminant of PMU Inks

In three surveys, we have identified 79 bacterial isolates from a total of 76 PMU ink samples. They included 50 isolates from 29 PMU inks in the previous two surveys [14,15] and 29 isolates from 47 PMU inks in the present survey. To systematically investigate patterns of species occurrence and species–species co-occurrences, the 79 bacterial isolates were mapped to produce a species-centric co-occurrence network (76-PP SCN) with 49 nodes (bacterial species) and 165 edges (co-occurrence relationships) (Figure 2). As shown in Figure 2, the 76-PP SCN consists of two subnetworks, an SCN (29-PMU SCN) from the previous surveys of 29 PMU inks, which contains 35 nodes (35 species belonging to eight genera) connected by 104 co-occurrence edges and an SCN (47-PMU SCN) from the present survey of 47 PMU inks, which contains 21 nodes (21 species belonging to nine genera) and 69 edges. As revealed in the Venn diagram (Figure 2), the two subnetworks exhibited 28 and 14 exclusive nodes (or noncore species), respectively, and 7 shared nodes (core species) and, at the genus level, five and six noncore genera, respectively, and three core genera (Figure 2B). Conclusively, 49 bacterial species have been identified as bacterial contaminants of PMU inks, and, among them, 14 species belonging to six genera were newly identified in the present survey.

#### 3.3.2. Three *Bacillus* spp., *B. pumilus*, *B. licheniformis*, and *B. cereus*, Showing the Highest Occurrence and Co-Occurrence Degree

As shown in the 47-PMU SCN (the present survey), among the 21 nodes, 12 species represented by red nodes are pathogenic (~51%). The three core pathogenic *Bacillus* spp., *B. pumilus*, *B. licheniformis*, and *B. cereus* [23,24,25], are the top three nodes with the highest occurrence of three, two, and two, respectively; they were detected in 4–6% of the PMU ink samples. On the other hand, the top three hub nodes showing a connection degree (i.e., no. of edges) of ≥12 were three *Bacillus* spp., a noncore species, *B. clausii*, and two core species, *B. cereus* and *B. firmus*. Interestingly, Sphingomonas paucimobilis, a noncore species of the 47-PMU SCN, was the only Gram-negative bacterium identified from all three PMU ink surveys. On the other hand, among the 35 nodes of the 29-PMU SCN from the previous survey, 11 species were pathogenic (~31%). The three core Bacillus spp., which showed the highest occurrence and degree in the 47-PMU SCN, also are the top three nodes in terms of occurrence and degree, except for the noncore bacterium B. thuringiensis with the highest connection degree of 12 in the 29-PMU SCN. Conclusively, the SCN of bacterial isolates from PMU inks is scale-free with apparent connection preferences, and a few isolates dominate the overall occurrence (avg. occurrence of 1.57) and connectivity (avg. degree of 5.94). Unlike the tattoo inks [14,15], however, no networks showed any statistical supporting relationship between the occurrence and connection degree (Pearson correlation: r = 0.13, *p* = 0.46 for 29-PMU SCN, r = 0.01, *p* = 0.98 for 47-PMU SCN, r = 0.11, *p* = 0.45 for 76-PP SCN).

#### 3.3.3. Seven Core Bacterial spp. Commonly Identified from Both the Previous and Present Surveys

Seven bacterial spp. (core species in the 76-PP SCN) were commonly identified in both the previous and present surveys (Figure 2). The seven core species included five pathogens and two nonpathogens. *Bacillus* spp. were dominant in the core species. Four core pathogenic *Bacillus* spp., *B. pumilus*, *B. licheniformis*, *B. cereus*, and *B. firmus*, showed a relatively high occurrence and degree. *B. pumilus* especially showed a strong co-occurrence with two other core pathogenic spp., *B. licheniformis* and *B. cereus* (Figure 2). The seven core Gram-positive bacterial spp., which are often known as extreme environment survivors [26], show a higher occurrence and connection degree than the noncore species, suggesting possible positive selection pressure of PMU inks and their resistance to the selection pressure in terms of microbial contamination.

Epistatic interactions are functional combinations of two or more contamination factors, including selection pressure of ink environments, microbial contamination sources and their degree of microbial complexity, and phylogenetic and/or phenotypic relationships among the contaminants [14]. These epistatic interactions appear to be neither simple nor random, and they seemed to determine the occurrence, co-occurrence, and diversity of microbial contaminants in PMU and tattoo inks. Therefore, it is a challenge to convert the microbial contamination data of the PMU ink surveys into practical and productive public health information via identification of contamination factors. A network-based analysis of the relationships among bacterial isolates seems to be essential to reduce the degree of epistatic complexity of contamination factors. As revealed in the 76-PP SCN, bacterial contaminants showed apparent connection preferences, which are not limited to pairwise co-occurrences. The relatively low average occurrence (1.57) but relatively high average degree (5.94) of the bacterial contaminants in the PMU SCNs could be explained by their relatively close phylogenetic/phenotypic–phenotypic relationships. In addition, the dominance and strong co-occurrence of spore-forming extreme environment survivors Bacillus spp. in the core species suggest the existence of positive selection and their ubiquity in nature, relatively high contamination, and resistance to the selection pressure in PMU inks.

## 4. Conclusions

While several studies report high contamination rates in sealed tattoo inks and describe the putative relevance for public health [10,11,12,13,27], information is lacking on the microbial contamination of PMU inks. This present study indicates that unopened and sealed PMU inks contain microorganisms, many of which are pathogens. Contaminated inks may lead to human microbial infections if injected into the dermis. As revealed in the species-centric co-occurrence network of diverse bacterial contaminants, a variety of contamination factors/routes could be responsible for the high degree of contamination and microbial diversity of PMU inks. On the other hand, a high degree of occurrence and coexistence of GP spores or endospore-forming *Bacillus*-centric contaminants with a close phylogenetic/phenotypic relationship suggests selection pressure and bacterial resistance in PMU inks. Nevertheless, it is also clear that more survey data should be added to identify all possible contamination factors/routes to prevent microbial contamination in PMU inks. 

## Figures and Tables

**Figure 1 microorganisms-10-00820-f001:**
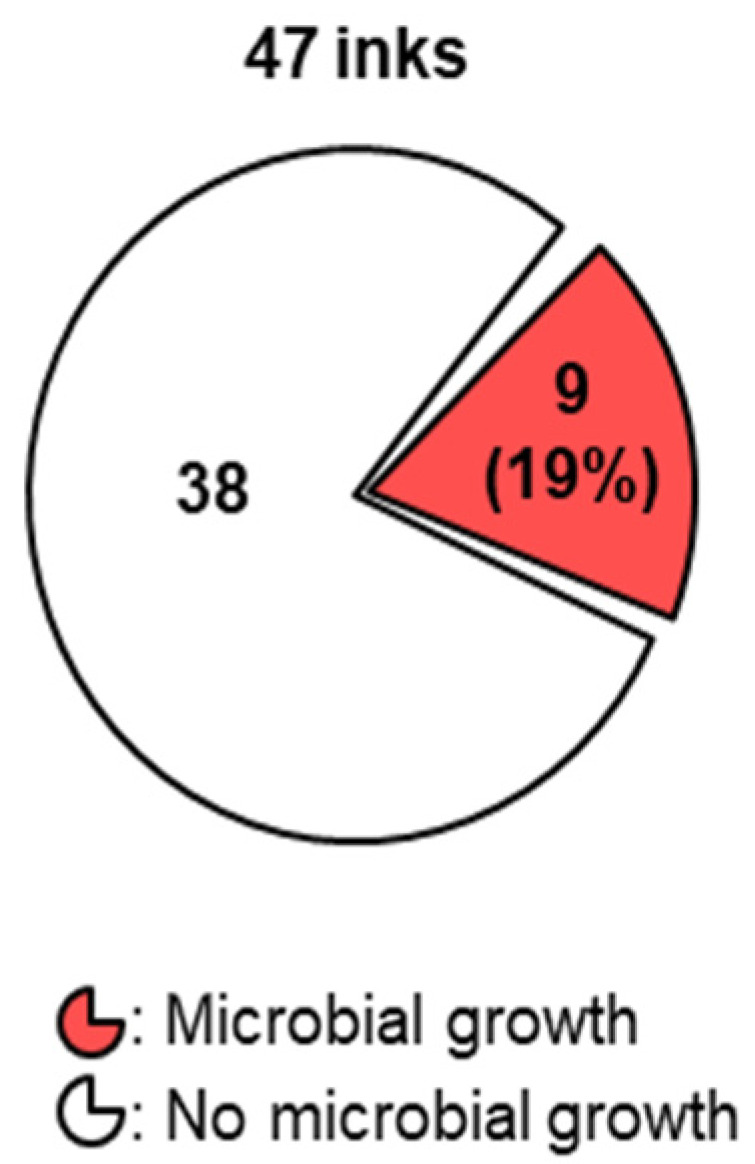
Microbial content of PMU inks. A total of 9 out of 47 PMU inks contained microorganisms.

**Figure 2 microorganisms-10-00820-f002:**
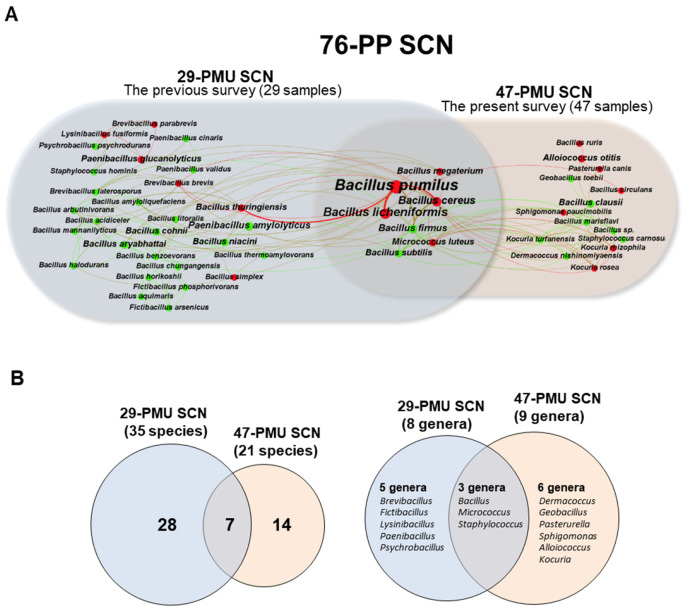
Comparative species-centric co-occurrence network analysis of the bacterial isolates from the previous and present PMU ink surveys. (**A**), a species-centric co-occurrence network (76-PP SCN) of the 76 PMU ink samples (29 samples of the previous survey and 47 samples of the present survey). In the network, color and size of the nodes indicate pathogenicity (red, pathogenic and green, nonpathogenic) and occurrence rate of the bacterial isolates from PMU ink samples, respectively. Thickness of the edges was weighted by co-occurrence rate. (**B**), Venn diagrams of the two subnetworks, 29-PMU SCN (the previous survey) and 47-PMU SCN (the present survey), at the species (**left**) and the genus level (**right**), respectively.

**Table 1 microorganisms-10-00820-t001:** Detection and identification of microorganisms in PMU inks.

Ink #	Manufacturer	Ink Type	Claim Sterility	CFU/g	Identification
1	1	Mb	Y	980	*Staphylococcus carnosus*
2		Mb	Y	<10	
3		Mb	NA	<10	
4		Mb	NA	<10	
5		PMU	NA	<10	
6		PMU	NA	<10	
7	2	Mb	NA	<10	
8		Mb	NA	<10	*Sphingomonas paucimobilis*
9	3	Mb/PMU	Y	<250	*Bacillus ruris* *Alloiococcus otitis*
10		Mb/PMU	Y	<250	*Bacillus clausii* *Geobacillus toebii* *Alloiococcus otitis* *Pasteurella canis*
11	4	Mb	NA	<10	
12		Mb	NA	<10	
13		Mb	NA	<10	
14		PMU	NA	<10	
15		PMU	NA	<10	
16		PMU	NA	<10	
17	5	Mb/PMU	Y	<10	
18		Mb/PMU	Y	<10	
19		Mb/PMU	Y	<10	
20		Mb/PMU	Y	<10	
21		Mb/PMU	Y	<10	
22		Mb/PMU	Y	<10	
23		Mb/PMU	Y	<10	
24		Mb/PMU	Y	<10	
25	6	Mb	NA	<10	
26		Mb	NA	<10	
27		Mb	NA	<10	
28		PMU	NA	<10	
29		PMU	NA	<10	
30		PMU	NA	<10	
31	7	PMU	NA	<250	*Bacillus cereus* *Bacillus pumilus*
32		PMU	NA	<250	*Bacillus megaterium* *Bacillus pumilus*
33		PMU	NA	<250	*Bacillus circulans* *Bacillus licheniformis*
34		PMU	NA	<10	
35		PMU	NA	<250	*Bacillus licheniformis* *Bacillus pumilus*
36		PMU	NA	<10	
37	8	Mb/PMU	NA	<250	*Bacillus clausii* *Bacillus firmus* *Bacillus subtilis* *Kocuria turfanensis* *Bacillus marisflavi* *Bacillus simplex* *Dermacoccus nishinomiyaensis* *Kocuria rhizophila* *Kocuria rosea* *Micrococcus luteus*
38		Mb/PMU	NA	<10	
39		Mb/PMU	NA	<10	
40		PMU	NA	<10	
41		PMU	NA	<10	
42	9	Mb	Y	<10	
43		Mb	Y	<10	
44		Mb	Y	<10	
45		PMU	Y	<10	
46		PMU	Y	<10	
47		PMU	Y	<10	

## Data Availability

The data presented in this study are available in the article and Appendix A.

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
