# Peer review of "Microbiological Survey of 47 Permanent Makeup Inks Available in the United States"

_microorganisms, 2022, doi:10.3390/microorganisms10040820_

Round 1

Reviewer 1 Report

In their study " Microbiological Survey of 47 Permanent Makeup Inks Availa-2 ble in the United States "

the authors do an appropriate study with interesting microbial results.

Particularly well done is the detailed comparison to previous data (Fig. 2).

Comments:

Overall this is a nice and interesting article, however the data quality should be enhanced:

Methods section is fine and instructive,

but in the results section there should be the original / raw data measured be presented in supplement (e.g in excel sheets)

The results should be broken down detailing the results and data from the different methods used (plate tests, PCR and 16S rRNA...) and it would be good if also the respective dilution data (plate tests) and the PCR detection data would be  presented.

The data availability statement is not "not applicable", but the raw data should be there and currently this is lacking. this is critical for acceptance of the article in this reviewer´s opinion.

Finally, it would be good to have a proper little discussion section, not a mix of results and discussion, so sort that out a bit better

Author Response

Responses to comments and suggestions for authors from reviewer #1

The authors do an appropriate study with interesting microbial results. Particularly well done is the detailed comparison to previous data (Fig. 2).

Comments:

[Comments] Overall this is a nice and interesting article, however the data quality should be enhanced: Methods section is fine and instructive, but in the results section there should be the original/raw data measured be presented in supplement (e.g in excel sheets).

[Responses] As suggested, we have provided the original data in an Excel file.

[Comments] The results should be broken down detailing the results and data from the different methods used (plate tests, PCR and 16S rRNA...) and it would be good if also the respective dilution data (plate tests) and the PCR detection data would be presented.

[Responses] While we agree that breaking down the results and data will help understand the results in a little more detail, I am afraid that it is necessary. For example, CFU values in Table 1 is the rounded numbers of the average of plate counts from 3 dilutions from 10-1 to 10-3 as described in Materials and Method section, but it isn’t thought that we would need all the original numbers to understand the values and the whole study. The way and format that the results and data was provided in this manuscript is basically the same that we used in our previous two tattoo ink microbial survey papers.

[Comments] The data availability statement is not "not applicable", but the raw data should be there and currently this is lacking. this is critical for acceptance of the article in this reviewer´s opinion.

[Responses] We have corrected the statement to “The data presented in this study are available in the article and supplementary material”.

[Comments] Finally, it would be good to have a proper little discussion section, not a mix of results and discussion, so sort that out a bit better

[Responses] We agree that providing an article with the results and discussion as separate sections has its own advantages, but we would like to keep the format of this manuscript where the two sections are combined. Originally, our first tattoo ink survey paper, “Microbiological survey of commercial tattoo and permanent makeup inks available in the US (Nho et al. 2018)”, was published as the two sections separately, because we wanted to discuss holistically the issues of microbial contamination of commercial tattoo inks. We also wanted to give the readers a continuity in the discussion and so the readers can view and analyze the complete study in one go. However, our second tattoo ink survey paper “Microbial contamination of tattoo and permanent makeup inks marketed in the US: a follow-up study (Nho et al. 2020)”, was published as the two sections combined, because the second paper was “a follow-up study” where the issues of microbial contamination of tattoo inks were narrowed down to focus on a particular issue. In the second paper, we discussed results of the study immediately after presenting them, thus the readers wouldn’t have to switch between sections to find the discussion that correlates its results.

The current manuscript up for publication also only examines one of the issues, “microbial contamination of permanent makeup inks”, that was lacking in our first and second survey study. Therefore, we want to keep the format of the current manuscript as it is for the same reason. We want to discuss the results right next to it, thus readers would immediately understand each of the results without spending time to go back to the results section.

Reviewer 2 Report

Tattooing and permanent make-up have dramatically increased over the last years, with a concomitant increase of health risks for consumers (microbiological contamination and the presence of nanomaterials or toxic substances in the inks). The Authors of this paper investigated the level of microbial contamination of 47 permanent make-up inks. Their study  showed that a significant proportion of the inks examined were contaminated with micro-organisms, including pathogenic bacteria, an emerging safety concern for public health. The article is well organized and well written, and in my opinion it deserves to be published in Microorganisms in the present form.

Author Response

Responses to comments and suggestions for authors from reviewer #2

[Comments] Tattooing and permanent make-up have dramatically increased over the last years, with a concomitant increase of health risks for consumers (microbiological contamination and the presence of nanomaterials or toxic substances in the inks). The Authors of this paper investigated the level of microbial contamination of 47 permanent make-up inks. Their study showed that a significant proportion of the inks examined were contaminated with micro-organisms, including pathogenic bacteria, an emerging safety concern for public health. The article is well organized and well written, and in my opinion, it deserves to be published in Microorganisms in the present form.

[Responses] We appreciate your supportive comments.